# PRISMA Systematic Review of Electroencephalographic (EEG) Microstates as Biomarkers: Secondary Findings in Memory Functions

**DOI:** 10.3390/neurolint17100160

**Published:** 2025-10-04

**Authors:** Fernan Alexis Casas Osorio, Leonardo Juan Ramirez Lopez, Diego Renza Torres

**Affiliations:** 1Faculty of Medicine and Health Sciences, Universidad Militar Nueva Granada, Bogotá 110111481, Colombia; 2TIGUM Research Group, Universidad Militar Nueva Granda, Bogotá 110111481, Colombia; leonardo.ramirez@unimilitar.edu.co (L.J.R.L.);; 3Faculty of Engineering, Universidad Militar Nueva Granada, Bogotá 110111481, Colombia

**Keywords:** electroencephalography, EEG, microstates, biomarkers, memory

## Abstract

**Background**: Monitoring brain activity through electroencephalography (EEG) has led to significant advancements in the study of brain microstates and their relationship with cognitive processes, such as memory. **Objective**: A systematic literature review was conducted following the PRISMA methodology, with the aim of identifying and analyzing potential biomarkers of memory functions derived from EEG microstate analysis. **Methods**: Searches were performed in five major databases (PubMed, Scopus, Web of Science, Springer, and institutional registers), covering studies published between 2019 and 2024. The initial search retrieved 179 records; after removing duplicates and ineligible works, 18 full-text articles were evaluated. Finally, 10 original studies met the inclusion criteria. Although primarily focused on other pathologies or baseline conditions, these studies reported relevant findings related to memory processes. This allowed for an exploratory synthesis of the potential role of EEG microstates as indirect biomarkers of memory. **Results**: The findings revealed that microstates, particularly microstates C and D, show significant alterations in their duration, coverage, and occurrence in various pathologies, such as Alzheimer’s disease, schizophrenia, and attention disorders, highlighting their potential as noninvasive biomarkers. **Conclusions**: Although methodological variability across studies represents a limitation, this review provides a solid foundation for future research aimed at standardizing the use of EEG microstates in clinical applications, improving diagnostic accuracy in memory-related diseases. Overall, EEG microstates hold great promise in both neuroscientific research and clinical practice.

## 1. Introduction

The monitoring of brain activity has been significantly facilitated through electroencephalographic (EEG) techniques, which allow for the recording of electrical signals generated by the cerebral cortex, the outer surface of the brain [1,2,3,4]. The cerebral cortex is divided into five main areas, four of which (frontal, parietal, temporal, and occipital) are accessible through EEG recordings and are involved in cognitive, sensory, and motor functions, as well as associative processes [3,5,6]. These areas are highly interconnected, and their activity does not occur in isolation; rather, they participate in complex distributed networks that integrate both within and across lobes, depending on the cerebral hemisphere [7,8].

Neuroimaging techniques such as functional magnetic resonance imaging (fMRI) and quantitative EEG have been employed to explore brain functioning and its relationship with cognitive processes [9,10]. In this context, a significant correlation has been found between the formation of microstates and images based on neuronal metabolism [9,11,12,13]. EEG microstates are described as brief periods of stable brain activity lasting between 60 and 120 milliseconds, and they reflect the activation patterns of resting-state neural networks [9,10]. These microstates represent the temporal organization of brain activity into “processing blocks” that allow transitions between different functional states [9].

The concept of EEG microstates as the “atoms of thought”—the fundamental units of cognitive processing—was first introduced in early studies [10]. Each microstate is associated with a specific cortical activation configuration, reflecting the processing of different types of information [14]. Spatial cluster analysis, which groups EEG topographic maps based on spatial similarity, has been used to study the organization of microstates [15]. This approach employs algorithms such as k-means to identify the most representative brain topographies, generating classes or groups corresponding to different microstates [15,16].

Despite significant advances in the study of brain activity through electroencephalography (EEG), challenges remain in understanding and standardizing EEG microstates in relation to various cognitive functions, such as memory. Microstates, defined as brief and recurrent patterns of brain activation, have been proposed as potential biomarkers for a variety of cognitive processes and neurological pathologies, including memory impairments [11,17]. However, the existing literature reveals methodological variability in the identification and analysis of microstates, which complicates the integration of findings across studies [18,19]. This fragmentation has limited a comprehensive understanding of how microstates are specifically related to memory alterations, leaving a significant knowledge gap that hinders their clinical application as reliable biomarkers [20,21].

In this context, it is necessary to examine whether EEG microstates—traditionally studied in various clinical and physiological conditions [10,22]—can also provide insights into memory. Several reasons support this: (1) they capture millisecond-scale brain dynamics, enabling the study of fast processes such as encoding and retrieval; (2) they are linked to networks crucial for memory, including the default mode network (microstate C) and the frontoparietal network (microstate D); and (3) studies in schizophrenia, Alzheimer’s disease, and mild cognitive impairment have shown parameter alterations associated with memory deficits. Moreover, as non-invasive, reproducible, and cost-effective measures, microstates emerge as promising biomarkers, though validation in larger longitudinal cohorts is still required [23,24,25].

The present systematic review, conducted in accordance with the Preferred Reporting Items for Systematic reviews and Meta-Analyses (PRISMA) methodology [26], aims to address this question by identifying, classifying, and analyzing original studies published between 2019 and 2024 that employed EEG microstates as neurophysiological markers. Specifically, it seeks to answer the following research question: What evidence exists regarding the association between EEG microstates and memory functions in clinical or physiological studies where memory was not a primary outcome, and how might these findings support the use of microstates as exploratory biomarkers? This approach aims not only to map an underexplored area of the literature, but also to evaluate the potential of EEG microstates as complementary tools in cognitive neuroscience research and clinical contexts involving memory processes [27].

## 2. Methodology

### 2.1. Microstate Analysis

Microstates are determined based on the spatial correlation of recorded cortical activity, and their stability is assessed through measures such as Global Field Power (GFP), which calculates the magnitude of the electric field generated by neurons at a given moment [9,28]. In EEG microstate analysis, the signal recorded across multiple channels is interpreted as a sequence of instantaneous topographies of electrical potentials on the scalp. To detect moments with the highest signal-to-noise ratio (SNR), the *GFP* is calculated at each time point. This *GFP* value is obtained by computing the square root of the mean of the squared differences between the voltage at each electrode and the average voltage across all electrodes:(1)$$GFP\left(t\right)=\sqrt{\frac{\sum_{i=1}^{n}{\left(v_{i}\left(t\right)-\bar{v}\left(t\right)\right)}^{2}}{n}},$$

In this equation, *v_i (t)* represents the voltage measured at electrode *i* at time *t, v  (t)* is the average voltage across all electrodes at time t, and n is the total number of electrodes. The points at which the GFP curve reaches local maxima—that is, points where the GFP is greater than that of adjacent time points—correspond to moments of highest field intensity, indicating a better signal-to-noise ratio (SNR) [28].

The topography of the electric field tends to remain stable between two locals minimum of the GFP curve, making GFP peaks representative of nearby topographies in time. Therefore, simplifying the original EEG signal and focusing on these local GFP maxima is an effective technique for improving the signal-to-noise ratio (SNR) and reducing the amount of data to be analyzed. These topographic maps, obtained at the local GFP maxima, are the ones subjected to further analysis, as shown in Figure 1.

Global Map Dissimilarity (GMD) is another measure used to assess topographic differences between consecutive microstates, regardless of signal intensity (Custo et al., 2017; Khanna et al., 2014) [28,29]. Global Map Dissimilarity, or GMD, refers to a method for measuring the similarity between different EEG topographies (scalp potentials). *GMD* is a metric that helps compare two maps of brain activity and is defined in a way that is unaffected by signal intensity—only by the topography itself (Khanna et al., 2014; Michel and Koenig, 2018) [9,28]. The *GMD* is calculated using the following formula:(2)$$GMD=\frac{\left\|\frac{x_{n}}{{GFP}_{n}}-\frac{x_{n^{\prime }}}{{GFP}_{n^{\prime }}}\right\|}{\sqrt{C}}$$

In which, *x_n_ y x_n_*_′_ are the EEG maps at two different time points, *GFP_n_ y GFP_n_*_′_ are the *GFP* values at those moments, *C* is the number of electrodes or channels in the EEG.

The *GMD* has a value of 0 when two maps are identical, and a maximum value of 2 when the maps have inverted topographies (i.e., the signs of the values are reversed). This metric is crucial because it allows for the assessment of topographic changes in EEG maps regardless of signal intensity, focusing solely on how the signals vary spatially across the scalp [28].

### 2.2. Functional Associations of Microstates

The analysis of microstates has proven particularly useful in the study of psychiatric and neurological disorders. In schizophrenia, for example, alterations in the duration and frequency of microstates have been observed, suggesting that these patterns could serve as biomarkers for the diagnosis and monitoring of the disease [10,30]. Similarly, it has been identified that microstates have specific relationships with different cognitive functions: microstate A is associated with phonological processing, while microstate B is related to visual processing. Meanwhile, microstates C and D are linked to attention and autonomic processing [16,18].

Microstate A has been consistently linked with phonological and verbal processing. Studies have shown that this microstate has a topographical configuration with a focus of activation in the left occipital region towards the right frontal region, suggesting its participation in the integration of sensory information and language organization [14]. It has been observed that in patients with impairments in verbal processing, such as those with dyslexia, the duration of microstate A is significantly reduced, supporting its relationship with linguistic functions [20].

Microstate B is associated with visual processing. Its activation focuses on the occipital areas, especially in the right hemisphere towards the left frontal and temporal regions [16]. This microstate is predominantly activated during tasks requiring rapid and efficient visual processing, such as object identification and spatial navigation. In conditions such as visual agnosia, a reduction in the stability and frequency of microstate B has been observed, suggesting that its dysfunction could be associated with visual deficits [29]. Alterations in this microstate have been observed in patients with anxiety disorders and also a differentiation through microstates with mood disorders, indicating its relevance in autonomic processing and emotional regulation [31].

Microstate C has been linked with autonomic functions and attention. This microstate shows a topographical configuration centered in the anterior region and has been associated with the activation of the autonomic control network, suggesting its involvement in physiological regulation, such as heart rate and respiration [13,16]. Additionally, it has been linked with the ability to maintain sustained attention and with vigilance tasks [32].

Microstate D is strongly associated with oriented attention and the integration of eye movements. This microstate has shown predominantly central activation, involved in visuomotor coordination [10]. Its dysfunction has been correlated with difficulties in performing coordinated eye movements and spatial orientation. In patients with attention deficit hyperactivity disorder (ADHD), greater instability in this microstate has been reported, which may be related to the attentional difficulties characteristic of this condition [33].

EEG microstates have proven to be valuable tools in neuroscientific research, providing a detailed view of the brain’s functional organization in both resting states and during cognitive task execution [9]. Current research has linked each microstate with specific functions, paving the way for their use as biomarkers for various neuropsychiatric conditions. Through topographical segmentation and mathematical analysis of the duration, frequency, and coverage of these microstates, it is possible to identify alterations in brain dynamics that could be crucial for the diagnosis and treatment of disorders such as schizophrenia, ADHD, and anxiety disorders [14].

Although notable progress has been made in studying brain activity using electroencephalography (EEG), significant challenges persist in understanding and standardizing EEG microstates in relation to cognitive functions such as memory. Microstates, defined as brief and recurrent patterns of brain activation, have been proposed as potential biomarkers for a variety of cognitive processes and neurological pathologies, including memory impairments [11,17]. However, the existing literature showed methodological variations in the identification and analysis of microstates, making it difficult to integrate the results from different studies [18,19]. This fragmentation limited the comprehensive understanding of how microstates specifically relate to memory alterations, leaving a significant gap in knowledge that hindered their clinical application as reliable biomarkers [20,21].

In this context, the present study addressed these gaps through a systematic review based on the PRISMA methodology. The retrospective and standardized analysis of previous studies allowed for the identification of consistent EEG microstate patterns associated with memory impairments. By standardizing methodologies and synthesizing existing findings, a framework for the characterization of these biomarkers is consolidated, providing a significant value proposition for their clinical application in the early diagnosis and monitoring of memory disorders. This approach not only fills a key gap in the literature [10,22], but also lays the groundwork for future research and developments in the field of cognitive neuroscience, enabling a better understanding of brain dynamics in relation to memory [34,35].

### 2.3. Review of Research Articles

This systematic review follows the guidelines established by the PRISMA statement (Preferred Reporting Items for Systematic Reviews and Meta-Analyses), aiming to synthesize and evaluate the available evidence on the relationship between resting-state electroencephalographic (EEG) microstates and memory. To ensure the quality and reproducibility of the review, a rigorous approach was applied in the selection, extraction, and analysis of the included studies. The protocol for this review was registered on the OSF (Open Science Framework) platform, https://osf.io/36b48/?view_only=31d44691110c4a3f88109850afec6968 (accessed on 30 September 2025).

Specific eligibility criteria were defined to ensure the relevance of the selected studies. Experimental, observational, and cohort studies that assessed the relationship between resting-state EEG microstates and memory were included. Reviews, editorials, letters to the editor, and conference abstracts were not considered. The participants in the studies had to be human, with no age or gender restrictions. Additionally, only studies published in the last five years (2019–2024) and written in English or Spanish were included. Regarding the intervention, the selected studies had to measure and record resting-state EEG microstates and evaluate their correlation with memory. As comparators, studies involving subjects without any specific intervention related to EEG microstates were included.

The search strategy was designed to maximize sensitivity and specificity in identifying relevant studies. A systematic search was conducted in four electronic databases: Scopus, Web of Science (WOS), PubMed, and SpringerLink, between 2019 and 2024. Combinations of controlled terms (such as MeSH in PubMed) and specific keywords were used, including electroencephalography, EEG, microstates, biomarkers, and memory. The search algorithms were adapted to the syntax of each database (see Figure 2).

In Scopus: (electroencephalography OR EEG) AND (microstate* OR “EEG microstate*”) AND biomarker* AND memory in title, abstract, or keywords. In Web of Science: TS = (electroencephalography OR EEG) AND TS = (microstate* OR “EEG microstate*”) AND TS = (biomarker*) AND TS = (memory). In PubMed: ((“electroencephalography”[MeSH Terms] OR EEG[Title/Abstract]) AND (microstate*[Title/Abstract] OR “EEG microstate*”[Title/Abstract]) AND biomarker*[Title/Abstract] AND memory[All Fields]) with a date filter between 2019 and 2024. In SpringerLink: (“electroencephalography” OR EEG) AND memory AND (microstate OR “EEG microstates”) AND biomarker*. In addition, manual filters were applied to restrict the search to original studies, with access to the full text, and focused on the analysis of EEG microstates in adults (see Figure 2).

The selection process was developed in several phases. In the identification phase, all records obtained were imported into the Mendeley reference manager. Subsequently, studies were filtered based on titles and abstracts to exclude those that did not meet the eligibility criteria. In the eligibility phase, the full texts of potentially relevant studies were thoroughly reviewed. Any discrepancies between the reviewers were resolved through discussion, or if necessary, by a third reviewer. Finally, studies that met the established criteria were selected for inclusion.

For data extraction, a predefined form was used to collect relevant information from each study, including authors, publication year, country, participant characteristics (age and gender, among others), methods for evaluating EEG microstates and memory, primary and secondary outcomes, and any reported conflicts of interest. Quality assessment was performed by two reviewers, and any disagreements were resolved through discussion [37].

Data synthesis was conducted narratively, highlighting key features and main findings of the included studies. The results were reported following the PRISMA guidelines and included a flow diagram illustrating the study selection process, as well as detailed tables with the characteristics of the included studies and their results.

## 3. Results

In accordance with the PRISMA guidelines used for this systematic review, 18 documents were initially identified in the Scopus, Web of Science, Springer, and PubMed databases. After a detailed reading, two studies were excluded because they did not correspond to original research. Of the remaining 16, three were discarded because they had divergent objectives related to electroencephalographic recording, focusing on aspects unrelated to microstates or the search for biomarkers. Two studies conducted in pediatric populations and one additional study focused on motor functions were also excluded because they did not align with the specific objective of this review: to analyze electroencephalographic microstates as potential biomarkers in memory-related neurological diseases. As a result of this screening process, the final selection included 10 studies that met the established methodological and thematic criteria (see Table 1).

In the studies analyzed, the participants’ ages spanned a wide range, from 13 to 91 years, with an average age of 37.18 years. This range allows for a diverse representation of the population, covering both adolescents and older adults, which is crucial for assessing neurocognitive variations across development and aging. The variance of 132.89 reflects a high dispersion of ages, indicating that the studies include very different age groups. This age heterogeneity is essential for the generalization of results, as it enables the identification of consistent patterns across different life stages. The inclusion of both young participants (13 years old) and older adults (91 years old) facilitates understanding how EEG microstates may vary across the lifespan and in the presence of different cognitive and psychiatric conditions.

Discrepancies in EEG microstate findings can largely be attributed to three methodological factors: channel density, data preprocessing, and the type of cognitive assessment employed. Regarding channel density, low-resolution configurations (e.g., 19 channels) may compromise reliability and source localization, whereas high-density systems (e.g., 128 channels) yield more robust and detailed results. In preprocessing, the choice of frequency filters, artifact handling methods (ICA, ASR, or visual inspection), and the length of analyzed epochs directly influence parameters such as the duration, coverage, and occurrence of microstates. Cognitive tasks further condition interpretation: resting-state studies typically associate microstates with general brain networks (e.g., attentional or default mode), while specific tasks—such as visuospatial working memory—enable the identification of microstates linked to state or trait markers. Collectively, these methodological variations hinder direct comparison across studies and underscore the need for standardized protocols to establish microstates as reliable clinical biomarkers.

Six of the studies were conducted with young patients or middle-aged adults, while four other studies were conducted with elderly participants, up to 91 years old. Overall, the analyses were conducted on patients with medical histories, except for the study by Tait et al. [23] in the UK and San Marino, where the patients had amnesia or mild cognitive impairment (See Table 2).

The sample sizes in the studies varied significantly, ranging from 24 to 166 participants. This range of sample sizes indicates different methodological approaches, from detailed case studies to research with sufficient statistical power to detect moderate to large effects. Studies with larger sample sizes, such as those conducted by Shor et al. with 166 participants [38], allow for greater generalization of findings and superior statistical robustness. On the other hand, studies with smaller sample sizes may offer more detailed and specific analyses of subpopulations or particular conditions.

The studies also reveal variations in the gender distribution among participants. For example, the study by Tait et al. [23] reports a balanced gender distribution (52% male, 48% female), while other studies show a predominance of one gender over the other. This variability in gender distribution is relevant, as documented neurophysiological differences between women and men may influence the results of EEG studies. Gender analysis allows for a more nuanced understanding of how EEG microstates may differ between women and men, contributing to a more personalized approach in neuroscientific research [44,45].

On the other hand, several studies do not specify neurological histories and focus on general or at-risk populations, such as the study on schizophrenia and its endophenotypes [43]. This diversity in participant characteristics allows for a broad and detailed comparison of how EEG microstates can act as biomarkers in different neuropsychiatric conditions.

In the studies analyzed, the preprocessing steps for EEG data included several techniques to ensure the quality and consistency of the data before the analysis of microstates. Artifact removal was a common step in all the studies, implemented through specific algorithms such as the SARA algorithm from qEEG [27], or through independent component analysis (ICA), as applied in multiple studies [38,46].

Additionally, EEG signal filtering was crucial, with settings varying across studies, such as 2–17 Hz filtering study and 1–80 Hz filtering study. Notch filters, typically at 50 Hz, were also applied to remove line noise [27,47]. The interpolation of noisy channels and the average referencing were common additional steps to correct electrode-specific artifacts and provide a common reference. For example, Tait et al. used these methods as part of their preprocessing [23]. Some studies also included down sampling to reduce the amount of data and facilitate computational processing. While all the studies shared essential steps like artifact removal and signal filtering, the specific configurations and additional techniques varied, reflecting the particular needs and objectives of each research [41,48].

Similarly, in the reviewed studies, the equipment used and the configuration of the EEG channels varied significantly, showing the specific needs and objectives of each investigation. For example, the number of EEG channels used in the studies ranged from relatively simple configurations with 19 channels to more complex setups with 128 channels. Studies like those by Khoo et al. [27] and De Bock et al. [25] used 19-channel setups, which are suitable for microstate studies in general populations and allow for quicker, less invasive installations. In contrast, the study by Soni et al. [43] used 128 channels, providing much higher spatial resolution and allowing for a more detailed analysis of microstates and brain activity.

Electrode placement followed the international 10–20 system in most studies, as seen in the works of Khoo et al. [27] and Zhou et al. [46], among others. This system is widely used in EEG studies due to its standardization and effectiveness in capturing relevant brain signals. In studies with a higher number of channels, such as Soni et al. [43], electrode placement follows adaptations of the 10–20 system to accommodate the larger number of recording points.

Studies investigating specific clinical conditions, such as Alzheimer’s disease [23] and schizophrenia [43], in addition to memory impairments, tend to use a larger number of channels to better capture the complexity of brain activity associated with these conditions. A higher number of channels allows for greater spatial resolution and the ability to detect subtle patterns of brain activity that may be critical for the diagnosis and understanding of these diseases.

In studies comparing different participant groups, such as those with various psychiatric conditions [38] or different levels of risk for developing a disease [42], the EEG setup may vary to balance the need for detail with practicality and comfort for the participants. For example, the use of 19 channels may be sufficient to distinguish significant differences between groups without being too invasive or uncomfortable for the subjects.

Some studies implemented specific technologies to enhance the quality of EEG recordings. For example, the study by Chen et al. [39] used artifact removal techniques with the ASR algorithm and down sampling to 100 Hz to handle large amounts of data efficiently. These innovations are crucial to ensuring that the data obtained are of high quality and that the microstate analyses are accurate.

In the reviewed studies, the exposure times in milliseconds varied considerably depending on the specific experimental objectives. The study by Chen et al. [39] used 3500 ms exposures for the player and 1200 ms for the observer in a competitive game, capturing specific events of deceptive behavior. In contrast, the study by Tait et al. [23] employed much longer exposure times, using 20 s (20,000 ms) of resting-state EEG, which allowed for a more extensive assessment of brain microstates under static conditions. Most studies opted for exposure periods of several minutes during rest, such as the 6 min (360,000 ms) used by Khoo et al. [27] and Jatupornpoonsub et al. [41], which provided enough time to gather representative data of microstates under stable conditions. This variability in exposure times highlights the differences in study objectives, ranging from capturing rapid responses to specific stimuli to assessing general patterns of brain activity during rest.

In the analyzed studies, various scales were employed to assess memory, adapted to the specific objectives of each investigation. The Mini-Mental State Examination (MMSE) and the Rey Auditory Verbal Learning Test (RAVLT) were used in the study by Tait et al. [23] to assess cognitive decline and auditory verbal memory in patients with Alzheimer’s disease and mild cognitive impairment, providing a comprehensive assessment of cognitive functions and the ability to learn and retain verbal information.

On the other hand, the Balloon Analogue Risk Task (BART) was used in the study by Chen et al. [39], primarily designed to assess risk-taking behavior but also providing insights into working memory and decision-making under uncertainty. Additionally, the Brief Psychiatric Rating Scale (BPRS) was used in the studies by Zhou et al. [24] and De Bock et al. [25] to assess psychiatric symptoms, offering an indirect measure of affected cognitive functions.

In the study by Chu et al. [40], the Uniform Parkinson’s Disease Rating Scale-III (UPDRS-III) and the Montreal Cognitive Assessment (MoCA) were used to assess cognition in Parkinson’s patients. These scales, although focused on specific aspects of memory and cognition, significantly contribute to understanding the neurophysiological correlates observed in EEG microstates.

Similarly, in the studies reviewed, EEG microstates were found to play a role in understanding brain dynamics associated with different clinical and cognitive conditions. The findings related to each electroencephalographic microstate, as reported in the studies included in this systematic review, are described below, organized from the most frequently analyzed to the least addressed.

### 3.1. Microstates A and B

Microstates A and B have been mainly investigated in relation to psychiatric and cognitive conditions. De Bock et al. [25] reported a significant increase in the coverage and occurrence of microstate A in patients at risk or diagnosed with psychosis compared to healthy controls (HC), suggesting cortical hyperactivity associated with memory dysfunction and impaired cognitive processing. In contrast, microstate B showed a significant reduction in first-episode psychosis (FEP) patients compared to ultra-high-risk (UHR) individuals who did not transition to psychosis, pointing to potential alterations in executive and memory functions [25].

Soni et al. [43] identified significant differences in pre-trial and pre-response microstates between schizophrenia patients, their first-degree relatives, and controls, supporting the notion that microstates may act as endophenotypes reflecting genetic vulnerability and alterations in working memory and cognitive responsiveness.

Beyond schizophrenia, microstates A and B are linked to primary sensory networks—auditory/phonological and visual, respectively—and have been shown to be altered across multiple clinical and experimental contexts. In patients with end-stage renal disease (ESRD) at high risk of malnutrition-inflammation complex syndrome (MICS), both microstates showed increased coverage and occurrence [23], suggesting impaired sensory processing associated with cognitive decline. In FEP, while some studies found no significant differences using shared templates [24], others described increased parameters in microstate A and decreased parameters in microstate B, the latter proposed as a state marker of psychosis progression [25].

In visuospatial working memory (VSWM) tasks, the pre-trial Map 1—topographically similar to microstate A—appeared less frequently and with reduced coverage in schizophrenia patients, with its intracranial generator localized to the right inferior frontal gyrus (rIFG), a region critical for inhibitory control and behavioral organization [43]. Conversely, the pre-response Map 4—similar to microstate B—appeared less frequently in first-degree relatives than in controls, reinforcing its potential role as an endophenotypic marker of genetic predisposition to schizophrenia [43].

Taken together, although microstates A and B are not directly related to higher-order memory networks, their alterations in conditions such as ESRD, psychosis, and schizophrenia suggest that dysfunctions in basic auditory and visual processing may represent an early substrate of cognitive decline and memory impairment.

### 3.2. Microstate C

Microstate C has been analyzed in several of the studies included in this review. In healthy individuals, mild sleep deprivation has been shown to produce a significant increase in its mean duration, coverage, and occurrence, suggesting a direct impact on drowsiness, cognitive performance, and short-term memory [27]. Likewise, in patients with end-stage renal disease (ESRD), the parameters of microstate C (coverage, duration, and occurrence) showed a negative association with the severity of malnutrition-inflammation complex syndrome (MICS), indicating a possible link between the reduction in this microstate and the cognitive decline observed in this population [41].

From a functional perspective, microstate C is consistently associated with the salience network and the default mode network, particularly the posterior cingulate cortex and the precuneus [27]. These regions are essential for memory recollection and the allocation of attentional resources, and alterations in microstate C could therefore explain cognitive deficits observed under different conditions.

In sleep deprivation, the increase in the occurrence and coverage of microstate C reflects greater activity in default mode network regions that are vulnerable to lack of rest, which may represent a compensatory mechanism against cognitive deficits [27]. In ESRD, the progressive reduction in microstate C activity with increasing MICS severity suggests that dysfunction of the salience network could mediate cognitive decline in these patients [41].

In early Parkinson’s disease, a reduction in the temporal variability of brain regions associated with microstate C was found compared to healthy controls. Lower flexibility in frontal lobe connectivity within this network was correlated with poorer cognitive performance, as measured by the MoCA scale [40].

In psychosis and schizophrenia, results are heterogeneous: while some studies reported no significant differences [24,25], De Bock et al. and Luo et al. found an increased occurrence of microstate C in patients with schizophrenia [25,42]. This inconsistency could be explained by the possibility that, in the classical four-map model, microstate C may encompass two functionally distinct subcomponents, each associated with different resting-state networks [42].

Overall, microstate C emerges as a sensitive indicator of cognitive and mnemonic processes. Its increase following sleep deprivation may represent a compensatory mechanism, whereas its reduction in ESRD and its functional rigidity in Parkinson’s disease are associated with greater cognitive decline.

### 3.3. Microstate D

Microstate D has shown significant alterations across various clinical studies. Tait et al. [23] reported relevant topographic changes, with a longer mean duration in patients with Alzheimer’s disease (AD), reflecting slower transitions between neural networks and less efficient information processing, consistent with cognitive decline. Zhou et al. [24] observed that in first-episode psychosis (FEP) patients, the duration of microstate D was significantly shorter compared to healthy controls, which was associated with deficits in memory and executive functions. Additionally, De Bock et al. [25] found decreased coverage and occurrence of microstate D in individuals at ultra-high risk (UHR) of psychosis, suggesting impairments in cognitive organization and memory in this group.

Functionally, microstate D is linked to attentional networks—dorsal, ventral, and frontoparietal—involved in working memory and selective attention [27,30,42,43]. In AD, in addition to altered topography, reduced activation of parietal regions was observed [23], consistent with frontoparietal dysfunction and the memory/attention deficits characteristic of the disease. Moreover, an overall increase in microstate duration was reported, indicating less flexible and more repetitive processing [23].

Under physiological conditions such as mild sleep deprivation, a significant increase in the occurrence of microstate D was found [27], interpreted as a compensatory mechanism in which the attentional network remains active to counteract drowsiness and maintain homeostasis.

In early Parkinson’s disease (PD), spatial variability of the microstate D network showed significant alterations: reduced variability in the left temporal lobe was associated with better cognitive performance (MoCA), while reduced variability in the right occipital lobe correlated with worse motor symptoms (UPDRS-III) [40].

Overall, microstate D stands out as a robust indicator of attentional network and working memory integrity. Its alterations are evident across multiple disorders: parietal dysfunction and increased duration in AD, compensatory increases after sleep deprivation, reduced stability in schizophrenia and psychosis risk, and region-specific inflexibility in PD. These findings consolidate microstate D as a promising biomarker for understanding and assessing memory and attention deficits in diverse neuropsychiatric conditions.

The findings of these studies underscore the importance of EEG microstates as biomarkers for various cognitive and psychiatric conditions. The observed changes in the duration, coverage, and occurrence of microstates C, D, A, and B correlate with alterations in memory and other cognitive functions, providing a valuable tool for the assessment and monitoring of these conditions.

## 4. Discussion

The analyzed studies showed that microstates could capture real-time brain dynamics, providing valuable insights into brain connectivity and cognitive functions—particularly memory, which is the focus of this review. EEG microstates allow continuous evaluation of brain activity in real time, which is especially useful for identifying patterns that can act as biomarkers in pathological states. Their non-invasiveness is a significant advantage, especially in clinical populations where other techniques may be more difficult to implement. Furthermore, microstates offer insights into the brain’s functional connectivity during resting states, a crucial aspect in diseases such as Alzheimer’s, schizophrenia, and other psychiatric disorders. The study by Shor et al. [38] emphasizes how the spatiotemporal configurations of microstates can distinguish between psychiatric disorders, thereby expanding the diagnostic utility of EEG microstates.

Microstate C, associated with memory and attention, has shown significant alterations in various pathological contexts. In the study by Khoo et al. [27], mild sleep deprivation led to an increase in the duration, coverage, and occurrence of microstate C, suggesting that this microstate could serve as a sensitive marker for short-term memory impairment, as there is evidence linking changes in this microstate to working memory. Similarly, Jatupornpoonsub et al. [41] found that microstate C was negatively correlated with cognitive function in patients with end-stage renal disease (ESRD), indicating a possible link between malnutrition, inflammation, and cognitive decline.

In the study by Luo et al. [42], microstate C showed reduced duration and coverage in patients with first-episode schizophrenia (FESZ), suggesting a dysfunction in information integration and working memory. Additionally, Soni et al. [43] observed significant differences in microstate C among patients with schizophrenia, their first-degree relatives, and healthy controls, highlighting its potential as an endophenotype for the disorder.

Lastly, the study by Shor et al. [38] reinforces the importance of microstate C in diagnostic differentiation between depression and schizophrenia, showing that this microstate can serve as a distinctive biomarker in psychiatric disorders.

Microstate D has been associated with information integration processes and showed significant changes across various diseases. Tait et al. [23] found that patients with Alzheimer’s presented a longer duration of microstate D, possibly reflecting difficulties in memory consolidation. In the study by Zhou et al. [24], patients with first-episode psychosis showed a shorter duration of microstate D, suggesting deficits in cognitive integration.

Chu et al. [40] observed lower spatial variability in microstate D in patients with Parkinson’s disease, which may be related to the characteristic motor and cognitive deficits. Furthermore, De Bock et al. [25] reported a decrease in the coverage and occurrence of microstate D in ultra-high-risk (UHR) individuals for transitioning to psychosis, indicating potential alterations in memory and cognitive organization. Shor et al. [38] also highlighted the relevance of microstate D in differentiating between depression and schizophrenia, showing that this microstate may reflect heterogeneity in neural connectivity in these conditions.

These microstates were primarily studied in relation to vigilance and executive function. De Bock et al. [25] observed an increase in the coverage and occurrence of microstate A in patients at high risk of transitioning to psychosis, which may reflect a state of hyper-alertness or anxiety that negatively impacts memory and attention. Microstate B showed decreased occurrence in patients with first-episode psychosis, according to Zhou et al. [24], suggesting a deficit in executive function and planning ability.

Chu et al. [40] also reported greater temporal variability in microstate B in patients with Parkinson’s, possibly related to cognitive and motor instability. Additionally, the study by Chen et al. [39] explored the use of microstate B in the context of deceptive behavior, demonstrating its relevance in evaluating decision-making and working memory under risk-related situations.

The reviewed studies indicate a strong correlation between EEG microstates—particularly microstates C and D—and changes in memory. In patients with Alzheimer’s, microstate D showed prolonged duration, potentially reflecting difficulty in consolidating new information [23]. In schizophrenia, alterations in the duration and coverage of microstates C and D suggest dysfunction in working memory and information processing capacity [42,43]. These observations are consistent across different studies, reinforcing the potential of EEG microstates as non-invasive biomarkers for the diagnosis and monitoring of memory disorders. Shor et al. [38] also emphasized how microstate configurations may be specific to different psychiatric disorders, underscoring their value in differential diagnosis.

Beyond memory, EEG microstates have shown important correlations with other cognitive domains. Microstate A, related to vigilance, and microstate B, linked to executive function, have been studied in contexts such as psychosis and Parkinson’s disease [25,40]. These microstates reflect alterations in vigilance and executive planning, which can directly impact memory and other cognitive functions. The study by Tait et al. [23], linking microstate D to cognitive decline in Alzheimer’s disease, is particularly revealing, demonstrating how microstates can reflect not only memory status but also the overall integrity of cognitive function. Additionally, research by Chen et al. [39] highlights how microstates can be used to evaluate decision-making and working memory in specific contexts such as deceptive behavior, further expanding their application as biomarkers.

On the other hand, it can also be inferred those microstates may have limitations in the study of memory and other pathologies due to interindividual variability and the need for extensive preprocessing—important challenges in using EEG microstates as biomarkers. EEG recording quality is also critical, as artifacts may complicate accurate microstate identification. Moreover, interpreting microstates in relation to specific cognitive functions still requires further validation through broader clinical studies. The methodological complexity, as seen in studies using advanced techniques such as Independent Component Analysis (ICA) and noisy channel interpolation [38,39], may be a barrier to broader clinical implementation.

Likewise, it would have been valuable to conduct a study including other neurological diseases beyond those related to memory, as mentioned in this review, which becomes a significant limitation in understanding the full spectrum of microstate utility.

For future studies, it is recommended to broaden research on the application of EEG microstates as biomarkers in various neurological and psychiatric diseases, focusing on clinical validation in larger and more diverse cohorts. Specifically, integrating EEG microstates with other neuroimaging techniques and biological markers is suggested to improve diagnostic and prognostic accuracy. It would also be valuable to investigate interindividual variability and factors influencing microstates, such as age, gender, and comorbidities, in order to develop more personalized predictive models.

In this review, we focused on EEG microstates as potential biomarkers of memory and cognition; however, we acknowledge that other complementary techniques may also be valuable. Recent evidence has shown that functional brain connectivity can serve as a highly sensitive marker for the early detection of cognitive decline in clinically healthy individuals carrying pathological cerebrospinal fluid biomarkers such as amyloid and tau, highlighting the importance of integrating multiple approaches [49]. Thus, while microstates provide a valuable window into the temporal dynamics of brain activity, their combination with connectivity measures may enhance the identification of robust biomarkers of memory processes.

Similarly, longitudinal studies could be designed to assess how changes in microstates over time correlate with disease progression and treatment response, which would allow EEG microstates to be used not only as diagnostic tools but also as dynamic indicators of clinical evolution.

### Limitations

The limitations of the systematic review are acknowledged, such as the heterogeneity between the studies, potential publication biases, and limitations in the methodological quality of the studies included. These limitations were discussed in depth to provide an appropriate context for the findings of the review. Conclusions were drawn based on the available evidence, and recommendations for future research around EEG microstates and memory were made.

## 5. Conclusions

EEG microstates are valuable tools for assessing brain activity in a variety of neurological and psychiatric conditions. Despite some limitations, such as interindividual variability and the need for thorough preprocessing, their ability to capture complex brain dynamics in real time makes them promising biomarkers. The findings of the reviewed studies support their use in clinical practice, particularly in the diagnosis and monitoring of memory disorders and other cognitive functions.

The diversity of diseases studied—from Alzheimer’s and Parkinson’s to schizophrenia and first-episode psychosis—as well as the differentiation between psychiatric disorders such as depression and schizophrenia, suggests that EEG microstates have a broad range of clinical applicability.

The available evidence indicates that the study of EEG microstates applied to memory still requires methodological standardization and broader validation. Future research should integrate both resting-state and task-based recordings, employ multimodal neuropsychological assessments, and implement harmonized acquisition and preprocessing protocols using high-density configurations, while also validating findings in low-density clinical systems. In addition, the use of advanced measures of complexity and transition dynamics, combined with machine learning approaches, may enhance diagnostic sensitivity. Finally, longitudinal studies, validation in independent cohorts, and the integration of multimodal biomarkers (EEG, neuroimaging, and plasma markers) represent essential steps to establish microstates as clinical biomarkers of memory and cognition.

Overall, the findings suggest that microstates A and B reflect alterations in auditory and visual processing, microstate C is associated with the salience network and episodic memory, and microstate D with attention and working memory, positioning them as promising indicators of cognitive processes, although their definitive validation requires further research.

## Figures and Tables

**Figure 1 neurolint-17-00160-f001:**
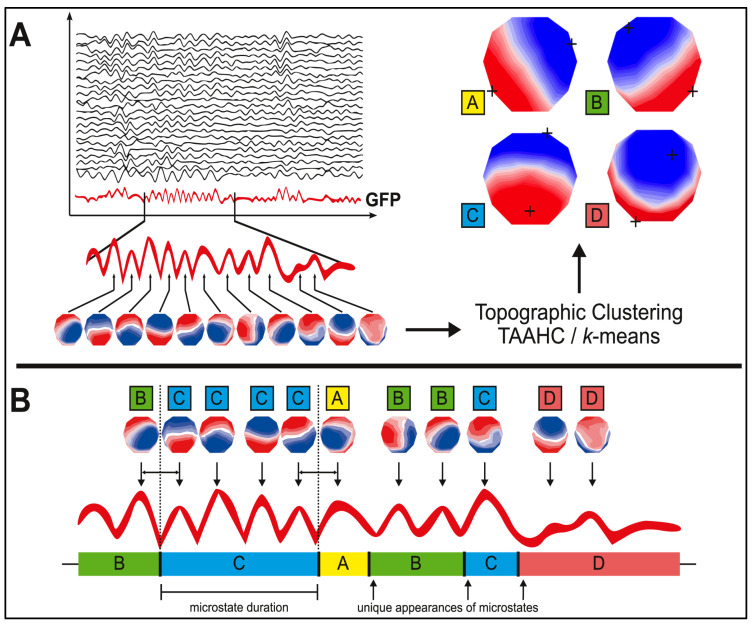
Detailed Method for Clustering and Analyzing Microstates in a Multichannel EEG Signal. A detailed method for clustering and analyzing microstates in a multichannel EEG signal is shown. At the top (**A**), the EEG signal recorded from multiple channels is displayed, which is filtered within a specific frequency band (usually between 8 and 12 Hz). Next, the global field power (GFP) curve—shown in red—is generated, measuring the intensity of the brain’s electric field over time. The peaks of this GFP curve indicate the moments of highest field intensity and, therefore, the best signal-to-noise ratio. At these peaks, the electrical potentials recorded by all electrodes are extracted, allowing for the construction of topographic maps. These maps are then subjected to a clustering algorithm that classifies them based on topographic similarity. Typically, studies on resting-state EEG microstates identify four main clustering maps (microstates), labeled A, B, C, and D. Finally (**B**), each topographic map extracted at the GFP peaks is assigned a label according to the microstate with which it most closely correlates. A particular microstate dominates for a period of 80 to 120 milliseconds before rapidly transitioning to another state [28].

**Figure 2 neurolint-17-00160-f002:**
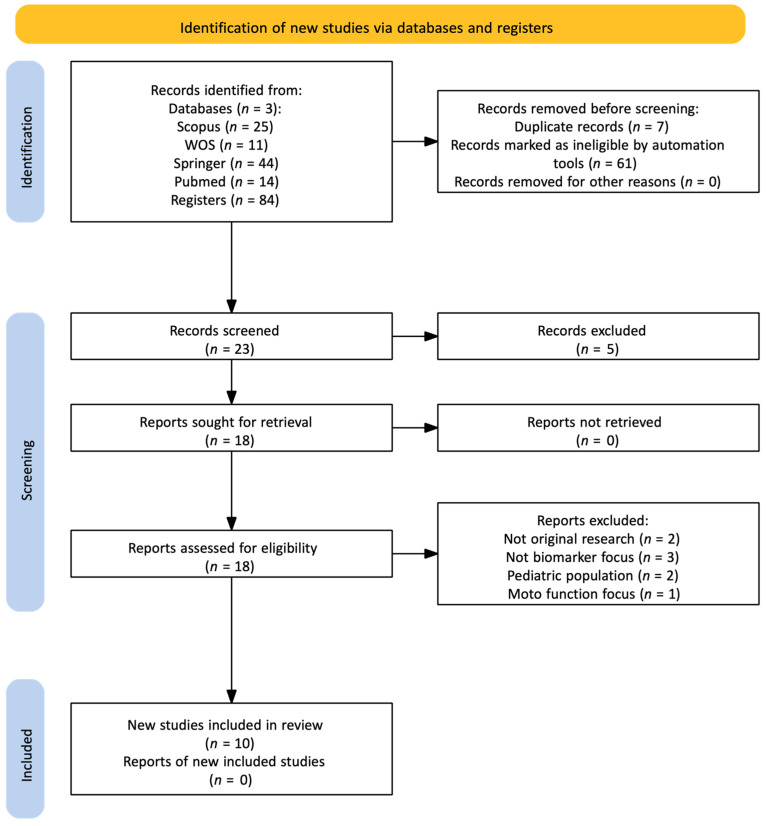
Flowchart of the studies included, following PRISMA guideline recommendations [36].

**Table 1 neurolint-17-00160-t001:** Demographic Data of the Participants.

Title	Age	Gender	Sample Size
Resting-state electroencephalography (EEG) microstates of healthy individuals following mild sleep deprivation [27]	21–40 years	66.7% women, 33.3% men	24 participants
EEG-based spatio-temporal relation signatures for the diagnosis of depression and schizophrenia [38]	18–91 years (media: 52.4 ± 18.7)	59.4% women	166 participants (96 controls, 28 with depression, 42 with schizophrenia)
EEG microstate complexity for aiding early diagnosis of Alzheimer’s disease [23]	18–91 years	52% men, 48% women	79 participants (21 AD, 25 MCI, 26 controls, 7 MCI)
An EEG dataset of neural signatures in a competitive two-player game encouraging deceptive behavior [39]	19–34 years (media: 25 ± 4.34)	50% men, 50% women	24 participants
Changes in oscillatory patterns of microstate sequence in patients with first-episode psychosis [24]	18–34 years (media: 22.8 ± 4.7)	70% men	142 participants (81 FEP, 61 controls)
Temporal and spatial variability of dynamic microstate brain network in early Parkinson’s disease [40]	62.4 ± 6.3 años (PD) 63.8 ± 5.5 años (HC)	31% hombres, 69% mujeres (PD) 50% hombres, 50% mujeres (HC)	51 participantes (29 PD, 22 HC)
EEG microstates as biomarker for psychosis in ultra-high-risk patients [25]	22.39 ± 5.24 years (HC), 25.32 ± 8.14 years (UHR-NT), 25.80 ± 7.20 years (UHR-T), 28.68 ± 7.64 years (FEP) 4o	12:13 (HC) 26:8 (UHR-NT) 11:9 (UHR-T) 19:10 (FEP)	108 participants (29 FEP, 20 UHR-T, 34 UHR-NT, 25 HC)
EEG Delta/Theta Ratio and microstate analysis originating novel biomarkers for malnutrition-inflammation complex syndrome in ESRD patients [41]	57.57 ± 14.88 years (mis ≤ 5), 59.13 ± 11.77 years (mis > 5)	69.6% women (mis ≤ 5), 65.2% women (mis > 5)	46 participants (23 mis ≤ 5, 23 mis > 5)
Biomarkers for prediction of schizophrenia: insights from resting-state EEG microstates [42]	13–40 years	Not especified	65 participants (20 FESZ, 19 UHR, 12 h, 14 HC)
Pre-trial and pre-response EEG microstates in schizophrenia: an endophenotypic marker [43]	21–40 years	66.7% women, 33.3% men	24 participants
**Conventions** **Age:** **HC: Healthy Controls** **UHR-NT: Ultra High-Risk Non-Transition** **UHR-T: Ultra High-Risk Transition.** **FEP: First Episode Psychosis.** **Gender: % women, % men: percentage of female and male participants in the study.**		**Medical Story** AD: Alzheimer’s Disease. MCI: Mild Cognitive Impairment. ESRD: End-Stage Renal Disease. PD: Parkinson’s Disease. **Source: own elaboration**	

**Table 2 neurolint-17-00160-t002:** Comparison of Methodologies.

Title	Preprocessing Steps	EEG Microstate Evaluation Methods	Memory Evaluation Methods	Summary of Results and Conclusions
Resting-state EEG microstates of healthy individuals following mild sleep deprivation [27]	Artifact removal (SARA), band-pass 2–17 Hz, average reference	Microstate analysis, 19 channels, 6 min resting EEG (10–20 system)	Karolinska Sleepiness Scale (Malay)	Mild sleep deprivation (>18 h) increased duration, coverage, and occurrence of microstate C and occurrence of D. C associated with DMN (precuneus, posterior cingulate) and D with attentional networks. Potential early markers of sleep deprivation effects.
EEG-based spatio-temporal relation signatures for the diagnosis of depression and schizophrenia [38]	Artifact removal (FASTER), high-pass 1 Hz, 50 Hz notch, interpolation of noisy channels, ICA	Dendrogram analysis, 19 channels, 500 s resting EEG (10–20 system)	Not specified	Did not use classical microstates; proposed dendrogram signature algorithm (PUDHS) differentiating depression, schizophrenia, and controls with high accuracy (AUC > 0.99). Objective tool for differential diagnosis.
EEG microstate complexity for aiding early diagnosis of Alzheimer’s disease [23]	Artifact removal, band-pass 1–40 Hz, interpolation of noisy channels, ICA	Microstate analysis, 64/19 channels, 20 s resting EEG (10–20 system)	MMSE, RAVLT	Microstate D altered in AD (reduced parietal activation). Lower Lempel-Ziv complexity and longer mean duration of microstates. EEG classifier achieved >80% sensitivity/specificity and predicted MCI conversion to AD.
An EEG dataset of neural signatures in a competitive two-player game encouraging deceptive behavior [39]	Downsampling 100 Hz, filters 1/49 Hz, ASR, interpolation, average reference, ICA	Microstate analysis, 31 channels, ERP-based (3500 ms player, 1200 ms observer)	Balloon Analogue Risk Task (BART)	Microstates applied to ERPs: differentiated instructed vs. spontaneous deception and player-observer outcomes. Increased GFP linked to P300. Useful for studying decision-making and deception.
Changes in oscillatory patterns of microstate sequence in first-episode psychosis [24]	Artifact removal (ICA), band-pass 1–80 Hz, 60 Hz notch, downsampling 100 Hz, average reference	Microstate analysis, 49 channels, 5 min resting EEG (10–20 system)	BPRS	Separate templates showed shorter A and D, and more frequent B and C in FEP. Introduced Chaos Game Representation (CGR), improving classification (AUC 0.61 vs. 0.46).
Temporal and spatial variability of dynamic microstate brain network in early Parkinson’s disease [40]	Artifact removal (ICA), band-pass 2–20 Hz, interpolation of noisy channels, ICA	Microstate analysis, 19 channels, 15–20 min resting EEG (10–20 system)	UPDRS-III, MoCA	Higher temporal variability of B and lower of C in early PD. Frontal variability (C) negatively correlated with MoCA. Spatial variability (D) linked to cognitive and motor symptoms. MCN-SVM classifier reached AUC 0.99.
EEG microstates as biomarker for psychosis in ultra-high-risk patients [25]	Artifact removal (ICA), band-pass 0.5–70 Hz, 50 Hz notch, interpolation, ICA	Microstate analysis, 19 channels, 8 min resting EEG (10–20 system)	BPRS	Microstate A ↑ in patients vs. controls; microstate B ↓ in FEP vs. UHR; microstate D ↓ in UHR-T vs. UHR-NT. A and B as state markers, D as trait marker predictive of psychosis transition.
EEG Delta/Theta ratio and microstate analysis in ESRD patients [41]	Artifact removal (ICA), band-pass 1–40 Hz, downsampling 128 Hz, Picard ICA	Microstate analysis, 19 channels, 6 min resting EEG (10–20 system)	MIS	ESRD patients with high MICS risk showed positive correlations with A/B, negative with C. Proposed MIC index combining A–C parameters, with 100% accuracy in discriminating high vs. low risk.
Biomarkers for prediction of schizophrenia: insights from resting-state EEG microstates [42]	Artifact removal (ICA), band-pass 1–80 Hz, 50 Hz notch, interpolation	Microstate analysis, 128 channels, 5 min resting EEG (10–20 system)	PANSS, CDSS, SIPS, MCCB	Six microstates (A–F) better explained data. Microstate D progressively decreased with schizophrenia severity. Random forest classifier with EEG + clinical tests reached 92% accuracy.
Pre-trial and pre-response EEG microstates in schizophrenia [43]	Band-pass 1–100 Hz, 50 Hz notch, artifact removal (ICA), average reference, downsampling 250 Hz	Microstate analysis, 128 channels, 50 ms pre-trial EEG, GFP map	Visuospatial working memory task	Map 1 (A-like) differentiated patients and controls (state marker); Map 4 (B-like) differentiated controls and relatives (trait marker). Generator localized in rIFG, key for inhibitory control and working memory.

**Abbreviations:** AD = Alzheimer’s Disease; AUC = Area Under the Curve; BPRS = Brief Psychiatric Rating Scale; CGR = Chaos Game Representation; DMN = Default Mode Network; MCI = Mild Cognitive Impairment (MCI); ERP = Event-Related Potential; ESRD = End-Stage Renal Disease; GFP = Global Field Power; ICA = Independent Component Analysis; MCCB = MATRICS Consensus Cognitive Battery; MIC = Microstate Index for MICS; MICS = Malnutrition-Inflammation Complex Syndrome; MMSE = Mini-Mental State Examination; MoCA = Montreal Cognitive Assessment; PANSS = Positive and Negative Syndrome Scale; PD = Parkinson’s Disease; PUDHS = Personalized Unsupervised Dendrogram Hierarchical Signature; RAVLT = Rey Auditory Verbal Learning Test; rIFG = right Inferior Frontal Gyrus; SARA = Semi-Automatic Artifact Rejection Algorithm; SIPS = Structured Interview for Prodromal Syndromes; SVM = Support Vector Machine; UPDRS-III = Unified Parkinson’s Disease Rating Scale part III; VSWM = Visuospatial Working Memory; UHR = Ultra-High Risk; UHR-T = UHR Transition; UHR-NT = UHR Non-Transition. **Source: own elaboration.**

## Data Availability

Not applicable.

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
