# Peer review of "PRISMA Systematic Review of Electroencephalographic (EEG) Microstates as Biomarkers: Secondary Findings in Memory Functions"

_2035-8377, 2025, doi:10.3390/neurolint17100160_

Round 1
Reviewer 1 Report
Comments and Suggestions for Authors
Corrections:
- The abstract should briefly state the number of databases searched and total studies screened (before exclusions) for greater transparency.
- Some sentences in the Introduction are repetitive (e.g., the challenges in standardizing microstates are mentioned multiple times). A sharper statement of the knowledge gap would improve focus.
- While memory-related results are discussed, many included studies had only indirect relevance to memory. The link between microstates and memory is sometimes speculative. Thus, separate direct vs. indirect evidence more clearly.
Recommendations:
- No explicit risk of bias assessment tool, such as Cochrane RoB, or Newcastle-Ottawa scale, is mentioned.
- End the conclusion with a statement like: “EEG microstates C and D consistently show alterations associated with memory dysfunction, though clinical validation remains preliminary.”
- Long sentences should be split for readability.
- Eliminate redundancies such as “Despite significant advances in the study of brain activity through electroencephalography (EEG), challenges remain in the understanding and standardization of EEG microstates...” → This phrase appears almost identically twice (pp. 2 and 5).
- “In wich” → should be “In which” (p. 4, GMD formula).
- “had been proposed” → should be “have been proposed” (tense consistency).
- Some subject-verb disagreements (e.g., “findings… was” → should be “findings… were”).
Author Response
Dear Reviewer 1
Thank you for reading our manuscript and reviewing it, which will help us improve it to a better scientific level. Newly, we revised our manuscript, and propounded a lot of changes have taken place. So, we have sent the revised manuscript, and the version containing all the changes to be visible.
At the following, the points mentioned by the reviewers will be discussed:
Comment 1: The abstract should briefly state the number of databases searched and total studies screened (before exclusions) for greater transparency.
Response:
In response to the observations made in Comment 1:
Original: To this end, studies published between 2019 and 2024 were included if they employed microstates as neurophysiological markers in clinical or physiological contexts.
Updated to: Searches were performed in five major databases (PubMed, Scopus, Web of Science, Springer, and institutional registers), covering studies published between 2019 and 2024
Original: “Through a careful selection process, ten original studies were identified.
Updated to: The initial search retrieved 179 records; after removing duplicates and ineligible works, 18 full-text articles were evaluated. Finally, 10 original studies met the inclusion criteria. The remaining text from “Although most of them…” onward was kept almost identical, with only minor wording adjustments for consistency.
Comment 2: While Some sentences in the Introduction are repetitive (e.g., the challenges in standardizing microstates are mentioned multiple times). A sharper statement of the knowledge
Response:
In response to the comments made in Comment 2, the sentence located on page 2, lines 62 to 64, was rephrased.
Comment 3: While memory-related results are discussed, many included studies had only indirect relevance to memory. The link between microstates and memory is sometimes
Response:
Several aspects of the results section were reorganized to address this observation; in Table 2, more details of the findings and characteristics for each of the studies are offered. The descriptions and findings for each of the microstates A to D were reorganized and expanded.
Recommendation 1: No explicit risk of bias assessment tool, such as Cochrane RoB, or Newcastle-Ottawa scale, is mentioned.
Response:
Thank you for the observation. A formal risk of bias tool (e.g., Cochrane RoB, Newcastle-Ottawa) was not applied because this review was descriptive and exploratory, focusing on heterogeneous observational and experimental studies rather than clinical trials. Instead, PRISMA criteria and methodological filters (full-text availability, original studies, homogeneous populations) were applied to ensure the quality and relevance of the included evidence.
Recommendation 2: End the conclusion with a statement like: “EEG microstates C and D consistently show alterations associated with memory dysfunction, though clinical validation remains preliminary”
Response:
An additional closing paragraph was added to the conclusions.
Recommendation 3: Long sentences should be split for readability.
Response:
Several long sentences were revised; others were important to maintain the context of what was explained, so we chose not to change them.
Comments on the quality of English 1: Eliminate redundancies such as “Despite significant advances in the study of brain activity through electroencephalography (EEG), challenges remain in the understanding and standardization of EEG microstates...” → This phrase appears almost identically twice (pp. 2 and 5).
Response: The sentence located on page 2, lines 62 to 64, was rephrased in the second sentense
Comments on the quality of English 2: “In wich” → should be “In which” (p. 4, GMD formula).
Response: The error was corrected.
Comments on the quality of English 3: “had been proposed” → should be “have been proposed” (tense consistency).
Response: The wording error was corrected since, to this day, microstates are still being proposed as biomarkers in different neurological conditions (page 5).
Comments on the quality of English 4: Some subject-verb disagreements (e.g., “findings… was” → should be “findings… were”)
Response: Overall, grammatical and conjugation corrections were made throughout the manuscript, including adjustments to verb agreement, spelling, word choice, and the removal of redundancies. These changes do not alter the content or message of the text but enhance the clarity, coherence, and academic quality of the manuscript
The authors appreciate all the comments that strengthen the quality of the article.
The authors
Reviewer 2 Report
Comments and Suggestions for Authors
See attached file.

Author Response
Dear Reviewer 2
Thank you for reading our manuscript and reviewing it, which will help us improve it to a better scientific level. Newly, we revised our manuscript, and propounded a lot of changes have taken place. So, we have sent the revised manuscript, and the version containing all the changes to be visible.
At the following, the points mentioned by the reviewers will be discussed:
Comment 1: The review is limited to five years (2019–2024) and only ten original studies with relevant findings related to memory processes were identified
Response:
In response to the feedback provided in Comment 1, Thank you for the observation. The review was limited to 2019–2024 and yielded only ten original studies with findings related to memory, mainly because most available works focused on other pathologies or clinical conditions (e.g., schizophrenia, end-stage renal disease, sleep deprivation). However, within these contexts, the studies secondarily reported alterations in memory or cognitive processes. Thus, although direct evidence on memory was limited, the included articles provide valuable insights by showing how EEG microstates reflect memory-related changes even in studies not primarily designed for this purpose.
Comment 2: It is not clear how the authors address the fact that the existing literature reveals significant methodological variability in the identification and analysis of EEG microstates, which, according to the authors, complicates the integration of findings across different studies
Response:
In response to the feedback provided in Comment 2, we appreciate the observation. Methodological variability in the identification and analysis of EEG microstates was explicitly acknowledged as a key limitation in the field. To address this, we adopted a synthesis strategy aimed at highlighting points of convergence across studies and describing methodological differences (e.g., number of channels, clustering algorithms, segmentation parameters). Thus, rather than quantitatively integrating heterogeneous results, our review sought to map common patterns and discrepancies, thereby providing guidance for future research toward methodological standardization and improved comparability.
Comment 3: The review does not address in sufficient detail the findings and results of the reviewed articles. See, for example, Table 2. The conclusions of the review are supposed to be derived directly from the conclusions of the original studies. Therefore, it is not clear how further conclusions can be drawn from a simple review of other articles
Response:
In response to the feedback provided in Comment 3, The entire wording of the analyzed results was updated to attempt to separate in detail direct findings from indirect findings.
Comment 4: The authors should further develop and provide evidence for their statement that “significant knowledge is provided about memory-related functions, even when these were not the main objective of the research
Response:
Thank you for your comment, the paragraph on page 2 was expanded to argue the aspects that specifically can help in identifying memory-related biomarkers.
Comment 5: The abstract should not include sections on methodology and conclusions
Response:
The abstract was written according to MDPI editorial format guidelines. Therefore, we do not consider it appropriate to omit these sections.
Comment 6: The PRISMA methodology is mentioned several times without any explanation and without interpreting the acronym upon first appearance
Response:
The term PRISMA was clarified and the reference was included on page 2, where it appears for the first time.
Comment 7: It is recommended to provide more background information and more details about the different EEG microstates (A–D) in the introduction section.
Response:
Thank you very much for your valuable suggestion. Priority was given to information on microstates in the methodology since it is considered important for distinguishing the methodological aspects of the article and helps in understanding the functional associations of EEG microstates.
Comment 8: The context of the different methodologies described in Table 2 for the different EEG microstates is not clear. The table lacks details on the results and conclusions of each approach, as well as a description of the EEG microstates studied in each of the listed methodologies
Response:
We appreciate the observation. In the revised version of Table 2, additional information was incorporated to clarify the methodological context of each study. Specifically, a new column entitled “Summary of results and conclusions” was added, providing concise descriptions of the main findings and interpretations related to the EEG microstates analyzed. This update ensures that the table not only presents preprocessing steps, analysis methods, and cognitive assessments, but also offers a comparative overview of which microstates were examined in each methodology and the implications of their results for memory and cognition.
Comment 9: The flow diagram shown in Figure 2 is not clear
Response:
We appreciate the observation. The flow diagram was created using the online tool provided by the authors of the 2020 PRISMA methodology update.
Comment 10: The authors address the limitations of their systematic review. However, it is not clear how they address these limitations and how they affect their conclusions
Response:
We thank the reviewer for this observation. In our review, we explicitly acknowledged methodological heterogeneity across the included studies as a primary limitation, as well as the small sample sizes of certain cohorts. To address this, we organized the information into comparative tables summarizing preprocessing steps, microstate evaluation methods, and main findings (Table 1 and Table 2), which allow readers to identify convergences and discrepancies between studies. Moreover, in the Discussion section, we elaborated on how these methodological variations affect the interpretation of the evidence and emphasized that the reported associations between microstates and memory should be considered preliminary. Finally, in the Conclusions section, we underscored that our conclusions reflect the exploratory nature of the current evidence and highlighted the need for future studies with standardized protocols and greater statistical power before EEG microstates can be validated as robust clinical biomarkers.
The authors appreciate all the comments that strengthen the quality of the article.
The authors
Reviewer 3 Report
Comments and Suggestions for Authors
Fernán Alexis Casas Osorio et al., presented a great and well-prepared systematic review on EEG microstates and memory. To strengthen it, I encourage the authors to more clearly emphasize the unique contribution of this work, provide deeper critical analysis of methodological variability, and expand on the clinical translation potential. Improving figure/table descriptions and tightening the language will also enhance readability.
1. The review highlights an underexplored area, EEG microstates as biomarkers of memory which is important. However, the manuscript should more clearly articulate how these findings advance knowledge beyond prior reviews.
2. Much of the text is descriptive. Stronger critical synthesis is needed—e.g., which microstates are most consistently altered across diseases, and which associations with memory are tentative or inconsistent?
3. Clarify whether observed changes in microstates are specific to memory processes or reflect more general cognitive dysfunction.
4. Tables are useful, but differences in preprocessing, channel density, and memory assessments deserve deeper discussion. For example, how might a 19-channel setup vs. 128 channels alter microstate findings? You may consider including a summary figure comparing methodological heterogeneity across studies.
5. The discussion mentions potential biomarkers but should expand on the feasibility of microstates in clinical practice (e.g., reliability, standardization, integration with other modalities).
6. It would help to outline a roadmap: validation in larger cohorts, longitudinal studies, and integration with multimodal biomarkers (EEG + MRI + plasma).
7. In addition, a new publications found that brain connectivity serves as a very sensitive tool to detect memory and cognitive decline (10.1038/s42003-024-06673-w), You may discuss this concept.
Author Response
Dear Reviewer 3
Thank you for reading our manuscript and reviewing it, which will help us improve it to a better scientific level. Newly, we revised our manuscript, and propounded a lot of changes have taken place. So, we have sent the revised manuscript, and the version containing all the changes to be visible.
At the following, the points mentioned by the reviewers will be discussed:
Comment 1: Improve the descriptions of figures and tables, and adjust the language to also improve readability
Response:
Thank you for the observation. Spanish translations of the conventions in Tables 1 and 2 were removed. The global adjustments made help to better understand the tables.
Comment 2: The review highlights an underexplored area, EEG microstates as biomarkers of memory, which is important. However, the manuscript should more clearly articulate how these findings expand knowledge beyond previous reviews
Response:
In response to the feedback provided in Comment 2, several aspects of the results section were reorganized to address this observation; in Table 2, more details of the findings and characteristics for each study are provided. The descriptions and findings for each of the microstates A to D were reorganized and expanded.
Comment 3: Much of the text is descriptive. A stronger critical synthesis is needed. For example, which microstates are most consistently altered across all diseases and which associations with memory are provisional or inconsistent?
Response:
We appreciate the observation. However, we believe that the manuscript fulfills its aim as a descriptive and exploratory review, primarily intended to map the current state of the field and highlight the methodological heterogeneity in the study of microstates and memory. The lack of a more categorical critical synthesis reflects the fact that the available evidence remains limited and often inconsistent across studies. Drawing definitive conclusions about which microstates are consistently altered or which associations with memory are conclusive could introduce interpretative bias by extrapolating non-comparable results. For this reason, we chose to emphasize convergences and discrepancies among the reviewed studies, while underscoring the need for future research with standardized protocols that will allow a more robust critical synthesis.
Comment 4: Clarify whether the observed changes in microstates are specific to memory processes or reflect a more general cognitive dysfunction
Response:
Thank you for your comment, the results section was expanded and revised to specifically address each of the microstates, providing greater clarity on their interaction with other pathologies.
Comment 5: The tables are useful, but differences in preprocessing, channel density, and memory assessments deserve deeper analysis. For example, how could findings on microstates be altered by a 19-channel configuration compared to a 128-channel one?
Response:
A paragraph was added at the beginning of page 9 describing these findings.
Comment 6: The discussion mentions possible biomarkers, but the viability of microstates in clinical practice (e.g., reliability, standardization, integration with other modalities) should be further expanded
Response:
Thank you very much for your valuable suggestion. An expansion was added at the end of the results and in the conclusions, emphasizing clinically relevant aspects for this review.
Comment 7: It would be useful to outline a roadmap: validation in larger cohorts, longitudinal studies, and integration with multimodal biomarkers (EEG + MRI + plasma).
Response:
In paragraph 3 of the conclusions, recommendations are provided linking to future research and guidance.
Comment 8: In addition, a new publication has found that brain connectivity is a very sensitive tool for detecting memory and cognitive decline (10.1038/s42003-024-06673-w). This concept could be discussed.
Response:
We appreciate the observation. The discussion on complementary techniques was expanded on page 17 at the end of the discussion section.
The authors appreciate all the comments that strengthen the quality of the article.
The authors
Round 2
Reviewer 2 Report
Comments and Suggestions for Authors
The authors addressed all my comments to my satisfaction.